# Mining Misconceptions in Mathematics

**Bryan Constantine Sadihin**    **Hector Rodriguez Rodriguez**    **Matteo Jiahao Chen**
Department of Computer Science
Tsinghua University
{wangwd24,lad24,chenjiah24}@mails.tsinghua.edu.cn

## 1  Introduction

Multiple-choice questions are widely used to evaluate student knowledge. This work proposes a model to predict the misconceptions associated with incorrect math answers. Well-designed MCQs need a set of correct answers and incorrect answers, commonly known as distractors. Distractors align with common misconceptions and serve as tools to identify knowledge gaps. However, determining the relationship between distractors and misconceptions is a hard and time-consuming task. Automatically performing the misconception mining would increase the efficiency of the multiple-choice question assessment.

There are two major challenges in understanding misconceptions. Firstly, high-quality questions include such a wide variety of misconceptions that they cannot be conventionally classified. Secondly, large language models (LLMs) struggle to understand misconceptions because they are optimized to produce correct answers and do not reason in the same manner as humans when answering questions.

We propose the use of an optimized LLM to analyze the incorrect answers and determine the associated misconception. Afterwards, the vector embedding of the LLM answer can be compared to the vector embedding of the established misconception categories to produce a list of suitable misconceptions.

## 2  Definition

**Evaluation Metric.**   We evaluate the model using the Mean Average Precision at 25 (MAP@25), a popular metric in information retrieval. It is calculated as follows:

$$MAP@25 = \frac{1}{U} \sum_{u=1}^{U} \sum_{k=1}^{\min(n,25)} P(k) \times rel(k) \tag{1}$$

where $U$ is the total number of observations, $P(k)$ is the precision at cutoff $k$, $n$ is the number of predictions per observation, and $rel(k)$ is an indicator function. The indicator function is 1 if the item at rank $k$ is the correct label, 0 otherwise. Once the correct label has been found, the subsequent predictions are disregarded.

## 3  Related work

### 3.1  Large Language Models for Math World Problems

Math Word Problems (MWPs) are mathematical exercises presented as written descriptions rather than direct equations [1]. LLMs must understand the relevant mathematical reasoning before formulating equations to solve the given problem. MWPs can be divided into three categories:

- **Question-Answer**: Each math question is only paired with the answer. SAT-Math [2] provides a high-school SAT Math dataset with multiple-choice questions.

Machine Learning (80245013-0) Proposal 2024.

- **Question-Equation-Answer**: Each math question is paired with the answer and the solving equation. The datasets target elementary school-level difficulty, and they are available in English (SVAMP[3], ParaMAWPS [4]) and in Chinese (Math23K [5], CM17K [6]).
- **Question-Rationale-Answer**: Each math question is paired with the answer and the reasoning path, akin to the Chain-of-Thought method, which includes the reasoning steps for correct problem-solving guidance [7]. Rationale data is generated using program induction (AQUA [8]) or LLMs (SAT-Math-COT [9])

LLMs such as Phi-3.5 [10] have demonstrated good performance in zero-shot MWP inferences. Additionally, fine-tuning an LLM with a math-specific dataset can improve the reasoning capabilities [11]. In-context learning with advanced prompting has shown promising results at a lower cost [1]. For example, an advanced prompting technique called "Self-Consistency" [12] significantly improved Chain-of-Thought by selecting the most consistent answer out of multiple LLM reasoning paths.

### 3.2 Vector embeddings

Vector embeddings are used in natural language processing for information retrieval [13]. One of the most popular embedding search alternatives is dense retrieval, which compares the similarities between the embedded texts to select information. The most common similarity metrics are the Euclidean distance, the cosine similarity, and the dot product similarity. Since the Euclidean distance and the dot product similarity are sensitive to both the magnitude and the direction, they can be useful for comparing embeddings that include measures. On the contrary, the cosine similarity is independent of the magnitude and can be used to compare the overall context [14].

Recent advancements in vector embeddings for information retrieval are not only limited to dense retrieval. BGE-M3 Embedding [15] uses an integrated relevance score that combines dense retrieval with sparse and multi-vector retrieval.

## 4 Proposed methods

Our main dataset is the Kaggle Eedi misconception dataset [16]. This dataset includes 1,868 English MWPs. Each question has one correct answer and three distractors that are aligned with one of the 2,586 possible misconceptions.

**Baseline**   LLMs such as Phi-3.5 have a strong zero-shot capability for mathematical reasoning. Therefore, they provide a straightforward baseline using zero-shot inference to propose incorrect answer misconceptions. However, the context window of the LLM may not fit the Eedi misconception list. To address this, we will narrow down the misconception list fed to the LLM by using misconceptions' vector embedding search to retrieve correlated misconceptions.

**In-context learning and fine-tuning**   Our proposal focuses on extending in-context learning and fine-tuning approaches for LLM mathematical reasoning to understand mathematical misconceptions. We aim to evaluate the reasoning ability difference between the in-context learning and the fine-tuning approach.

**Dataset generation and external datasets**   Some misconceptions rarely appear in the Eedi dataset distractors. We propose using other LLMs to balance the main dataset by generating misconception-aligned distractors. Additionally, we propose to extend the dataset using external Question-Rationale-Answer data to fine-tune the LLM for accurate distractor construction using answer reasoning.

**Multi-vector embedding retrieval**   There is a high degree of similarity between the misconceptions that characterize the answers to the same question. Thus, we propose the use of multi-vector embedding retrieval to leverage the similarity of the incorrect answers and their respective misconceptions. We will evaluate BGE-M3 performance to compute the vector embeddings of the misconception proposed by the LLM and the list of possible misconceptions. Additionally, other vector embedding techniques and similarity metrics could be explored.

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
