# OpenReview forum: "[Proposal-ML] Mining Misconception in Mathematics"
_tsinghua.edu.cn/THU/2024/Fall/AML — THU 2024 Fall AML Submission_

### Official Review · ~Daniel_Wang4 · 2024-11-06
**Mining Misconception in Mathematics Proposal Review**

**Rating:** 9
**Confidence:** 3

**Review:**

The proposal did an excellent job in describing the objective: to improve misconception detection in math assessments by leveraging the distractor answers to identify where exactly students are struggling with. This way of tracking misunderstandings is logical and well-founded.

The choice by the authors to employ large language models with vector embeddings and utilize the MAP@25 metric shows thoughtful planningd. However, relying only on the Kaggle Eedi dataset, while a good starting point, is too narrow to cover all the variation in misconceptions that exist. It would have been beneficial to extend the dataset with more variant examples and further widen the scope of the present study.

It would also strengthen the proposal to specify how misconceptions will be categorized, as similar types of misunderstandings could complicate the retrieval approach. Additional explanation of the choice of multi-vector embedding and anticipated challenges in in-context learning and fine-tuning would enhance the technical clarity.

Overall, this is a great proposal.

---

### Official Review · ~Ziyi_Liu9 · 2024-11-07
**Well Done**

**Rating:** 9
**Confidence:** 4

**Review:**

The introduction and proposed method are well-described, and there is a thorough review of the related work. However, Chapter 2 could be strengthened by including a mathematical formulation or a more detailed problem statement to clarify the research objectives.

---

### Official Review · ~Yunghwei_Lai1 · 2024-11-08

**Rating:** 10
**Confidence:** 5

**Review:**

The plan to use vector databases for retrieval-augmented generation is forward-thinking and highly beneficial, as it allows for efficient contextual storage and retrieval. This is especially valuable for reducing token-based costs in large language models, making the project both technically efficient and economically viable.

This project stands out for its innovative approaches to enhancing mathematical reasoning in LLMs, with a well-rounded dataset strategy, a focus on misconception modeling, and advanced retrieval techniques. Some additional clarity on implementation and evaluation would make the project even stronger, but overall, this is a promising and well-conceived proposal.

---

### Official Review · ~Kangping_Xu1 · 2024-11-09
**Review of "Mining Misconception in Mathematics"**

**Rating:** 9
**Confidence:** 2

**Review:**

## Pros:
- Practical value - Automating misconception identification could significantly reduce the time and effort needed to create effective multiple-choice math assessments, which currently require manual analysis of distractors.
- Comprehensive methodology - The proposal combines multiple advanced techniques (LLMs, in-context learning, fine-tuning, and multi-vector embeddings) to address the complexity of mathematical misconception analysis.

## Cons:
- Dataset limitations - The Eedi dataset only contains 1,868 problems, which may be insufficient for training robust models, especially given the large number of possible misconceptions (2,586). While they propose generating additional data, synthetic data might not accurately represent real student misconceptions.

---

### Official Review · ~Shuangyue_Geng1 · 2024-11-11
**Well-structured though definition could be clearer**

**Rating:** 9
**Confidence:** 3

**Review:**

This proposal is well-structured, presenting a novel method for categorizing misconceptions in math. The approach shows strong potential, and the organization is logical, with clear sections on related works and methodology. However, the second section's definitions are not easily accessible to those without domain knowledge, making it difficult to see their connection to the proposed method. Providing more detailed and clear explanations of the definition would enhance the proposal’s accessibility and impact.

---

### Official Review · ~Yida_Lu1 · 2024-11-11
**Clear task definition with method needed to be further clarified**

**Rating:** 8
**Confidence:** 3

**Review:**

This study utilizes an LLM to understand misconceptions within incorrect answers of math word problem through in-context learning and fine-tuning, aiming to identify the knowledge gaps of students. The task is meaningful in practice and clearly defined, and the proposal is well-structured.

However, the method part can be further clarified by illustrating how in-context learning and fine-tuning will be implemented in detail, as well as the innovation points within these two methods. Providing more implementation details will help readers better understand the methods proposed in this study.

---

### Official Review · ~Yu_Zhang61 · 2024-11-11
**Review of "Mining Misconception in Mathematics"**

**Rating:** 7
**Confidence:** 4

**Review:**

The "Mining Misconception in Mathematics" proposal outlines a novel approach for improving large language models' (LLMs) capacity to classify and understand mathematical misconceptions. By leveraging a combination of zero-shot inference, in-context learning, and fine-tuning, the authors aim to advance LLMs in diagnosing misconceptions from multiple-choice questions, where distractors are often tied to common errors. The model will employ a multi-vector embedding retrieval system to categorize misconceptions into a remarkably granular set of 2,586 categories, which could significantly improve educational assessments by providing detailed insights into students’ misunderstandings. This is an ambitious and potentially impactful project, though it would benefit from a clearer description of how the fine-tuning and in-context learning components will interact with multi-vector retrieval in practice. Additionally, the proposal could elaborate on how mean average precision will be used to interpret the model's performance meaningfully. Addressing these areas would improve the proposal's clarity, feasibility, and contribution to educational technology.

---

### Official Review · ~XueZeng1 · 2024-11-11

**Rating:** 8
**Confidence:** 4

**Review:**

It proposes to retrieve correlated misconceptions through vector embedding search of misconceptions, thereby narrowing down the list fed to the LLM.This is a clever way that enables the LLM to handle the misconception information related to the given dataset more effectively, avoiding processing difficulties or inaccuracies caused by the limitation of the context window.It combines different learning and fine-tuning methods.Focus is placed on expanding the in-context learning and fine-tuning methods for LLM's mathematical reasoning to understand mathematical misconceptions, and aiming to evaluate the difference in reasoning ability between these two methods.

Howerver,although it is mentioned in the text to evaluate the difference in reasoning ability of the LLM under different methods, the performance of BGE-M3, etc., overall, there is a lack of a specific and quantifiable evaluation indicator system

---

### Official Review · ~Changsong_Lei2 · 2024-11-12
**review of "Mining Misconception in Mathematics"**

**Rating:** 9
**Confidence:** 4

**Review:**

### Summary:
the proposal presents a model that leverages large language models (LLMs) to identify misconceptions in math multiple-choice questions (MCQs). By examining incorrect answers (distractors), the model aims to automatically predict the associated misconception.

### Pros:
- Proposes clear methodologies like multi-vector retrieval and in-context learning, showing a well-structured approach to addressing specific challenges in misconception mining.
- Gives a clear defination to the problem and the related dataset.

### Cons:
- Lacking a detailed discussion on the feasibility of this method. For instance, with fewer than 2000 MWPs in the dataset, can the model be effectively fine-tuned?

---

### Official Review · ~Yifan_Luo2 · 2024-11-12
**A chanllenging topic**

**Rating:** 8
**Confidence:** 4

**Review:**

**Summary:**

The proposal, titled "Mining Misconceptions in Mathematics," outlines a project aimed at using large language models (LLMs) to predict math misconceptions associated with incorrect multiple-choice answers. Traditional methods of identifying these misconceptions are time-consuming, so the authors propose an optimized LLM approach. They plan to use vector embeddings to match incorrect answers with misconception categories efficiently.

**Pros:**

1. **Efficiency:** Automating misconception identification can save time and resources.
2. **Innovative Use of LLMs:** Utilizing LLMs for this purpose can enhance their application in educational assessments.
3. **Data Utilization:** The approach makes good use of existing datasets and attempts to improve them by generating additional data.
4. **Advanced Techniques:** Incorporating vector embeddings and multi-vector retrieval can improve prediction accuracy.

**Cons:**

1. **Complexity:** Implementing such advanced techniques may require significant computational resources and expertise.
2. **LLM Limitations:** LLMs may struggle with nuances of misconceptions, as they are optimized for correct answers.
3. **Dataset Dependence:** The proposal relies heavily on the quality and comprehensiveness of the Eedi dataset.
4. **Scalability Concerns:** The approach may face challenges when scaling to diverse educational contexts or languages.

---

### Official Review · ~Chumeng_Jiang1 · 2024-11-12
**Needs More Clarification**

**Rating:** 7
**Confidence:** 3

**Review:**

This proposal aims to develop a model that can automatically identify misconceptions in mathematical multiple-choice questions. The authors propose in-context learning and fine-tuning techniques to enhance the LLM’s reasoning ability to associate incorrect answers with specific misconceptions. They also plan to leverage LLMs to generate additional data for underrepresented misconceptions.

**Strengths:**
- **Practical application value of the research topic:** Proper classification of misconceptions in multiple-choice questions is highly desired in educational institutions.
- **Detailed pipeline design:** The proposal involves multiple stages, with considerations for data generation, retrieval, fine-tuning, and evaluation.

**Weaknesses:**
- **Requires further clarification:** Why use multi-vector retrieval, and at what stage of the LLM pipeline will it be applied?
- **How to ensure the accuracy of data generated by other LLMs:** The proposal mentions that part of the dataset will be generated by other LLMs. This raises the question of whether training on this data aligns the model with the generating LLM’s patterns rather than with correct misconceptions.

---

### Official Review · ~Yangchi_Gao1 · 2024-11-12

**Rating:** 9
**Confidence:** 4

**Review:**

The proposal presents a promising approach to leveraging LLMs for educational assessment. It has the potential to significantly improve how misconceptions are identified and addressed in mathematics education.

Advantages:
1.The proposal to utilize large language models (LLMs) for mining misconceptions in mathematics is a novel approach that could significantly enhance educational assessment tools.
2.By focusing on identifying misconceptions associated with incorrect answers, the project has the potential to improve understanding of common student errors and knowledge gaps.

Disadvantage：
1.Misconceptions in mathematics can be highly nuanced and context-dependent, which might be challenging for an LLM to accurately capture.